# In Vitro Metabolism of Donepezil in Liver Microsomes Using Non-Targeted Metabolomics

**DOI:** 10.3390/pharmaceutics13070936

**Published:** 2021-06-23

**Authors:** Sin-Eun Kim, Hyung-Ju Seo, Yeojin Jeong, Gyung-Min Lee, Seung-Bae Ji, So-Young Park, Zhexue Wu, Sangkyu Lee, Sunghwan Kim, Kwang-Hyeon Liu

**Affiliations:** 1BK21 FOUR KNU Community-Based Intelligent Novel Drug Discovery Education Unit, College of Pharmacy and Research Institute of Pharmaceutical Sciences, Kyungpook National University, Daegu 41566, Korea; hjkopsty@gmail.com (S.-E.K.); hlhl103@naver.com (H.-J.S.); duwls9902@gmail.com (Y.J.); lgm00179@naver.com (G.-M.L.); wltmdqo2377@naver.com (S.-B.J.); soyoung561@hanmail.net (S.-Y.P.); sangkyu@knu.ac.kr (S.L.); 2Mass Spectrometry Based Convergence Research Institute, Kyungpook National University, Daegu 41566, Korea; wuzhexue@knu.ac.kr (Z.W.); sunghwank@knu.ac.kr (S.K.); 3Department of Chemistry, Kyungpook National University, Daegu 41566, Korea

**Keywords:** donepezil, metabolism, metabolomics, molecular networking, multivariate analysis

## Abstract

Donepezil is a reversible acetylcholinesterase inhibitor that is currently the most commonly prescribed drug for the treatment of Alzheimer’s disease. In general, donepezil is known as a safe and well-tolerated drug, and it was not associated with liver abnormalities in several clinical trials. However, rare cases of drug-related liver toxicity have been reported since it has become commercially available. Few studies have investigated the metabolic profile of donepezil, and the mechanism of liver damage caused by donepezil has not been elucidated. In this study, the in vitro metabolism of donepezil was investigated using liquid chromatography–tandem mass spectrometry based on a non-targeted metabolomics approach. To identify metabolites, the data were subjected to multivariate data analysis and molecular networking. A total of 21 donepezil metabolites (17 in human liver microsomes, 21 in mice liver microsomes, and 17 in rat liver microsomes) were detected including 14 newly identified metabolites. One potential reactive metabolite was identified in rat liver microsomal incubation samples. Metabolites were formed through four major metabolic pathways: (1) *O*-demethylation, (2) hydroxylation, (3) *N*-oxidation, and (4) *N*-debenzylation. This study indicates that a non-targeted metabolomics approach combined with molecular networking is a reliable tool to identify and detect unknown drug metabolites.

## 1. Introduction

Drug metabolism makes it easier for the body to remove drugs by making them more hydrophilic [1]. Metabolism consists of Phase I and Phase II reactions. Phase I reactions are oxidation, reduction, and hydrolysis, which are primarily catalyzed by cytochrome P450 (P450) enzymes. The Phase II reactions consist of conjugating molecules, such as glucuronic acid, sulfate, glutathione, amino acids, methyl groups, and acetyl groups, to the compound itself or metabolites generated from the introduction of polar functional groups into the parent compound through Phase I reactions [2]. In general, the pharmacological activity of most drugs is lost through metabolism. However, metabolism can still produce active metabolites and, in some cases, reactive or toxic metabolites that may affect drug safety [3,4,5,6]. Therefore, it is important to identify all of the drug metabolites in biological samples, such as urine, feces, and liver microsomes, which contain xenobiotic-metabolizing enzymes.

One method previously used in drug metabolism research is to filter the mass values of expected metabolites based on known metabolic reactions and analyze them by liquid chromatography–tandem mass spectrometry (LC-MS/MS). The chemical structure of the metabolites is then predicted by comparing the mass fragmentation pattern of the parent drug with that of a potential metabolite [7]. However, this method relies on the researcher’s knowledge of drug metabolism and the interpretation of mass fragmentation patterns in mass spectra. In addition, it is time-consuming, and it is not easy to detect a biotransformation that occurs through an unexpected pathway, such as structural rearrangement. Metabolomics can comprehensively analyze small molecules present in biological samples such as urine, plasma, and tissue [8,9]. Recently, metabolomics combining high-resolution mass spectrometry (MS) with multivariate data analysis has been used in drug metabolism studies and for the development of new biomarkers for disease diagnosis [10,11,12]. This new technique has been applied to determine the metabolic profile of amiodarone [13], isoniazid [14], itraconazole [15], ritonavir [16], and tolcapone [17] and has enabled the discovery of several new metabolites. In addition, molecular networking, which is currently used in various fields, such as natural product analysis, drug discovery, and drug metabolism, has been used as a tool for metabolite identification [18,19]. Molecular networking creates a network of compounds with structural similarities based on tandem MS data [18,19]. In general, drug metabolites contain the primary backbone of the parent drug and exhibit a similar fragmentation pattern in the MS/MS spectrum [18]. It is possible to identify structurally related metabolites through molecular networking based on the MS/MS spectrum of the parent drug [18,19]. Therefore, this technology presents an efficient tool for detecting drug metabolites [19].

Donepezil (Figure 1) is a reversible acetylcholinesterase inhibitor used to treat Alzheimer’s disease symptoms in many countries. Donepezil enhances cognitive performance by preserving normal levels of acetylcholine, a neurotransmitter that plays an important role in memory and cognitive function in the brain. [20,21] According to previous studies, donepezil is primarily metabolized by CYP3A4 and CYP2D6 to form *O*-demethylation, hydroxylation, *N*-oxidation, and *N*-dealkylation products [22,23]. These donepezil metabolites are further combined with glucuronic acid and excreted through the urine and bile [24,25]. The main adverse effects of donepezil include nausea, vomiting, and diarrhea, which are associated with cholinergic effects [26,27]. In addition, bradycardia, arrhythmia, and heart block have been reported in patients diagnosed with heart disease [28]. While donepezil has shown a favorable side effect profile in several clinical trials, rare cases of drug-related liver injury have been documented post-marketing [29,30,31]. Two of these studies reported the potential toxicity of the interactions between donepezil and a selective serotonin reuptake inhibitor [29,30,31]. However, the mechanism of donepezil-related toxicity is not well understood. Furthermore, the biotransformation of donepezil has not been thoroughly investigated.

In this study, we used a non-targeted metabolomics approach combined with molecular networking to comprehensively analyze the metabolism of donepezil in human, mouse, and rat liver microsomes. Additionally, as part of a study on the mechanism of liver damage induced by drugs, the production of reactive metabolites, which is considered the main cause of idiosyncratic side effects was investigated using glutathione (GSH) and potassium cyanide as trapping agents.

## 2. Materials and Methods 

### 2.1. Materials

5-*O*-Desmethyldonepezil, 6-*O*-desmethyldonepezil, hydroxydonepezil, 3-hydroxydonepezil, donepezil *N*-oxide, and desbenzyldonepezil were supplied by Toronto Research Chemicals (Figure 1, Toronto, ON, Canada). Donepezil hydrochloride was purchased from Tokyo Chemical Industry (Tokyo, Japan). Potassium cyanide (KCN) and reduced L-glutathione (GSH) were obtained from Sigma-Aldrich (St. Louis, MO, USA). Nicotinamide adenine dinucleotide phosphate reduced form (β-NADPH) was purchased from Oriental Yeast (Tokyo, Japan). Human liver microsomes (HLM (cytochrome P450, 0.411 nmol/mg protein; cytochrome b_5_, 0.449 nmol/mg protein; and NADPH-cytochrome c reductase, 182 ± 11 nmol/mg protein/min), catalog No. H2630), mouse liver microsomes (MLM (cytochrome P450, 0.939 nmol/mg protein; cytochrome b_5_, 0.414 nmol/mg protein; and NADPH-cytochrome c reductase, 138 ± 4 nmol/mg protein/min), catalog No. M1000), and rat liver microsomes (RLM (cytochrome P450, 0.646 nmol/mg protein; cytochrome b_5_, 0.484 nmol/mg protein; and NADPH-cytochrome c reductase, 124 ± 8 nmol/mg protein/min), catalog No. R1000) were acquired from XenoTech (Lenexa, KS, USA). All solvents for liquid chromatography-MS were obtained from Fisher Scientific Co. (Pittsburgh, PA, USA) with the highest grade commercially available.

### 2.2. Metabolite Profiling in Liver Microsomes 

Liver microsomal incubation samples were prepared by a modified method as described previously [32]. The stock solution of donepezil and β-NADPH were prepared in dimethylsulfoxide and water, respectively. Incubation mixtures were prepared with 50 µM donepezil and 1 mg/mL microsomal proteins in 100 mM potassium phosphate buffer (pH 7.4, 400 µL). After pre-incubation at 37 °C for 5 min, 20 mM β-NADPH (final concentration of 1.0 mM) was added to start the reaction. After incubation at 37 °C for 1 h, the reaction was terminated by adding an equal volume of cold methanol. Following centrifugation at 12,700 rpm at 4 °C for 10 min, the supernatant was evaporated using a Speed-Vac (Labconco, Kansas City, MO, USA). The residue was reconstituted in 100 µL of methanol, filtered through a 0.2-µm pore size syringe filter, and injected into the LC-MS/MS system for analysis. Controls were incubated without β-NADPH and all incubations were performed in triplicate.

### 2.3. Reactive Metabolite Profiling in Liver Microsomes

The soft electrophiles and iminium ions were captured by GSH and KCN, respectively. Incubation mixtures were prepared with 50 μM donepezil, 1 mg/mL microsomal proteins, and GSH (2.5 mM) or KCN (1.0 mM) in 100 mM potassium phosphate buffer (pH 7.4, 400 μL). After pre-incubation at 37 °C for 5 min, 20 mM β-NADPH (final concentration of 1.0 mM) was added to start the reaction. After incubation at 37 °C for 1 h, the reaction was terminated by adding an equal volume of cold methanol. Following centrifugation at 12,700 rpm, the supernatant was evaporated using a Speed-Vac. The residue was reconstituted in 100 μL of methanol, filtered, and injected into the LC-MS/MS system for analysis. All incubations were performed in triplicate.

### 2.4. LC-MS/MS Analysis

LC-MS/MS analysis was performed in the positive electrospray ionization (ESI) mode using a DIONEX UltiMate 3000 ultra-high performance liquid chromatography system coupled to a Q Exactive focus hybrid quadrupole-orbitrap mass spectrometer (Thermo Fisher Scientific Inc., Waltham, MA, USA). Chromatographic separation was performed on a Kinetex XB-C18 column (Phenomenex, Torrance, CA, USA, 100 × 2.1 mm, 2.6 μm, 100 Å), and the column oven temperature was maintained at 40 °C. The mobile phase consisted of solvent A (0.1% formic acid in water) and solvent B (0.1% formic acid in acetonitrile) with a flow rate of 200 μL/min. A gradient for the mobile phase was established as follows: 0–12 min, 10–35% solvent B; 12–15 min, 35–90% solvent B; 15–18 min, a re-equilibration time, 10% solvent B (total run time: 18 min). The following parameters were optimized for mass detection: heated capillary temperature: 320 °C; spray voltage: 3.5 kV; sheath gas flow rate: 35 arb; auxiliary gas flow rate: 12 arb; S-lens RF level: 50.0 V. The data were acquired in full scan and data-dependent MS/MS (ddMS^2^) mode. The parameters for the full scan mode were set as follows: resolution: 70,000; scan rage: 100–1000; AGC target: 1 × 10^6^; maximum injection time: 100 ms. For ddMS^2^ mode, resolution: 17,500; normalized collision energy: 30 eV, AGC target: 5 × 10^4^; maximum injection time 100 ms. The ddMS^2^ spectra were obtained for the three strongest peaks per cycle. All data were acquired and analyzed by using the Thermo Xcalibur 4.1 software.

### 2.5. Multivariate Analysis

Mass spectral data were processed with Compound Discoverer 2.1 (Thermo Fisher Scientific Inc.), which was used for peak detection, isotope grouping, chromatogram alignment, and generation of feature tables including molecular weight, retention time, and peak area values. The multivariate analysis based on the mass spectral data (*m/z* value, retention time, and peak intensity) was carried out using the SIMCA software (version 14.1, Umetrics AB, Umea, Sweden). Principal component analysis (PCA) and orthogonal partial least squares discriminant analysis (OPLS-DA) were conducted on Pareto-scaled data to identify significant differences between groups. The S-plot generated by OPLS-DA was used to assess the relative importance of variables in group separation. Compounds with *p* [1] > 0.01 in the S-plot was considered potential metabolites. The structure of the metabolite candidates was confirmed by comparing the MS/MS fragmentation pattern with that of the parent drug.

### 2.6. Molecular Networking

The molecular network was created by the Global Natural Products Social Molecular Networking (GNPS, http://gnps.ucsd.edu accessed on 20 May 2021) web platform. After converting the Thermo data file (.raw) to a .mzML file, they were processed with an MZmine open-source software, which was used for peak detection, chromatogram deconvolution, isotope grouping, and feature alignment. The results were analyzed using the feature networking workflow provided by GNPS. The parameters to create a network of donepezil were set as follows: precursor ion mass tolerance of 0.02 Da and fragment ion mass tolerance of 0.02 Da. The nodes are connected when the cosine score was greater than 0.5 and the MS/MS spectrum shared at least six matching peaks. Each node could be networked to the top 10 of the most related nodes.

## 3. Results and Discussion

### 3.1. Identification of In Vitro Phase I Donepezil Metabolites

In this study, we investigated the Phase I metabolism of donepezil using HLM, MLM, and RLM. A metabolomic and molecular networking approach were applied to comprehensively characterize donepezil’s metabolites. The results of a multivariate analysis results for ions produced by LC-MS/MS analysis of liver microsomal incubation samples incubated in the presence or absence of β-NADPH are shown in Figure 2. A PCA plot revealed that there was a clear difference between the sample and control. The S-plot generated from OPLS-DA was used to identify the variables contributing to this group separation. The top ions were identified as metabolites of donepezil and they are marked on the S-plot. A molecular network was also created using the GNPS web platform [33,34] for the data in the upper right of the S-plot (124 variables). A network containing 30 nodes was generated which were also considered potential drug metabolites (Figure 3). The structures of the metabolites were analyzed based on the accurate mass and fragment ion patterns, and the metabolites **M1**, **M2**, **M3b**, **M4**, and **M5** were confirmed by comparing with reference compounds. A total of 17, 21, and 17 metabolites were identified in HLM, MLM, and RLM, respectively. The reaction types, retention times, theoretical and measured masses, mass errors, chemical formulas, and fragment ions of donepezil and its metabolites are presented in Table 1. The mass error was within 5 ppm. Representative chromatograms and MS/MS spectra of the metabolites are shown in Figure 4 and Appendix A.

To identify the structure of the metabolites, we first determined the structural properties of the parent drug. The protonated molecular ion of donepezil (**M0**) was observed at *m/z* 380.2214 (mass error < 1.6 ppm) and eluted at 9.26 min. The MS/MS spectrum of donepezil by fragmenting *m/z* 380.2114 through collision gave the base peak at *m*/*z* 91.0546, which represents the dissociated benzylic carbocation [35] (Appendix A). Fragment ions were also generated at *m*/*z* 362.2105, 288.1591, 243.1378, and 151.0751, indicating the loss of a water molecule (−18 Da), the loss of a benzyl moiety (−92 Da), the loss of a water molecule and a benzyl methylamine group (−137 Da), and dimethoxybenzylic carbocation. The MS/MS spectrum of donepezil (Appendix A) was very similar to that of previous results [36].

The metabolites **M1** and **M2** were identified as 6-*O*- and 5-*O*-desmethyldonepezil by co-chromatography and the MS/MS spectral data of the authentic standards (Appendix A). The protonated molecular ions of metabolites **M1** and **M2** were observed at *m/z* 366.2064 (theoretical *m/z*) with mass errors of under 2.0 ppm, suggesting that the methyl group was removed from donepezil (−14 Da difference). They eluted at 7.64 and 7.85 min, respectively. Fragment ions were also generated at *m/z* 348.1958, 274.1438, 229.1223, and 137.0597 which were 14 Da smaller than those of donepezil, except for *m/z* 91.1, which was associated with the benzyl group. Characteristic fragment ions of desmethyldonepezil at *m/z* 91.1, 137.1, 229.1, 274.1, and 348.2 were also reported in previous studies [35,36,37]. The results indicate that demethylation occurred at the dimethoxyindanone moiety.

The protonated molecular ions of metabolite **M3a**, **M3b**, **M3c**, and **M4** were observed at *m/z* 396.2169 (theoretical *m/z*) and eluted at 6.85, 7.35, 7.59, and 9.98 min, respectively. The accurate mass measurements indicated that the chemical formula was C_24_H_29_NO_4_, indicating that one oxygen atom had been introduced into donepezil. The metabolite **M3a** produced fragment ions at *m/z* 91.0548, 151.0754, and 189.0918, indicating no metabolic changes in the benzyl group and dimethoxyindanone moiety (Appendix A). The other fragment ion of metabolite **M3a** was found at *m/z* 304.1552, suggesting that a hydroxylation occurred at the piperidine ring. Compared with the authentic reference compounds, metabolite **M3b** had a retention time and fragment ion pattern consistent with 3-hydroxydonepezil, indicating that hydroxylation occurred at the cyclopentanone moiety (Appendix A). The characteristic fragment ion of metabolite **M3c** was observed at *m/z* 107.0495, which is a 16-Da increase over the benzylic carbocation. The other major fragment ions that did not contain benzyl groups were the same as those of the parent drug at *m/z* 151.0758, 273.1475, and 290.1750 (Appendix A). This indicates that metabolite **M3c** underwent hydroxylation on the benzyl group. Examination of the MS/MS spectra indicated that the MS/MS spectrum of **M3c** could be readily distinguished from those of **M3a** and **M3b** because the formation of *m/z* 378.2064 was not favored when hydroxylation was on the aromatic ring (**M3c**). In general, a loss of water from metabolites with aliphatic hydroxylation was favored, whereas the loss of water was not favored when hydroxylation was phenolic [38]. The fragment ion peak produced from the loss of a water molecule was also not observed in the MS/MS spectrum of 3-hydroxydesloratadine, having a hydroxylated aromatic ring [38]. Based on these observations, **M3a** and **M3c** were tentatively identified as hydroxydonepezil in which oxygen was inserted on the piperidine and benzene rings of donepezil. However, the exact hydroxylation sites on the piperidine or benzene rings for metabolites **M3a** and **M3c** could not be determined.

The metabolite **M4** exhibited a protonated molecular ion [M+H]^+^ at *m/z* 396.2170 and gave a characteristic fragment ion at *m/z* 288.1592, 243.1389, 151.0752, and 91.0547, indicating that oxidation may occur at the nitrogen atom of the piperidine ring. This was confirmed by co-chromatography and the MS/MS spectral data of the authentic standards (Appendix A). Characteristic fragment ions of donepezil *N*-oxide at *m/z* 91.1, 151.1, 243.1, and 288.2 were also reported in previous studies [35,37]. Metabolite **M4** had a longer retention time compared with that of the parent drug, which is characteristic of *N*-oxide metabolite reported for several drugs [7,39].

The protonated molecular ion of metabolite **M5** was observed at *m/z* 290.1750 and eluted at 4.88 min. The fragment ions of metabolite **M5** were the same as those of the parent drug at *m/z* 272.1637, 205.0859, 189.0909, and 151.0752, but the fragment ion was absent at *m/z* 91.0542, related to the benzyl group (Appendix A). In addition, a fragment ion was observed at *m/z* 82.0657, which represents the dissociated tetrahydropyridine moiety. The MS/MS spectrum of **M5** indicated that a loss of the benzyl group occurred in the piperidine ring. The MS/MS spectrum of **M5** was very similar to that of previous studies [37]. This metabolite was also confirmed by comparing the retention time and fragment ion pattern of the authentic standard (Appendix A).

The protonated molecular ion of metabolite **M6** was observed at *m/z* 352.1905 (mass error < 1.0 ppm) and eluted at 6.33 min. The accurate mass measurements indicated that the chemical formula is C_22_H_25_NO_3_, suggesting that the two methyl groups were removed from donepezil (−28 Da difference). Metabolite **M6** had major fragment ions that were 28 Da smaller than those of donepezil at *m/z* 123.0441, 215.1069, 260.1281, and 334.1802 (Appendix A), suggesting that both methyl groups were lost in the dimethoxyindanone moiety of donepezil.

The protonated molecular ions of metabolite **M7a**, **M7b**, **M7c**, **M7d**, **M7e**, **M8a**, and **M8b** were observed at *m/z* 382.2013 (theoretical *m/z*) and eluted at 5.43, 5.74, 6.09, 6.49, 6.74, 8.39, and 8.69 min, respectively. The accurate mass measurements indicated that the chemical formula is C_23_H_27_NO_4_, indicating that one oxygen atom was introduced, and demethylation occurred. Metabolite **M7** had shorter retention times than the *O*-desmethyldonepezil (**M1** and **M2**), whereas metabolite **M8** had a longer retention time than the *O*-desmethyldonepezil, suggesting that mono-hydroxylation and *N*-oxidation occurred, respectively. Metabolite **M7a** and **M7b** had major fragment ions that were 14 Da smaller than those of hydroxydonepezil at *m/z* 272.1281, 290.1387, and 364.1907 (theoretical *m/z*, Appendix A), suggesting that one methyl group was removed from the dimethoxyindanone moiety of hydroxydonepezil. In addition, their fragment ions that were 16 Da larger than those of desmethyldonepezil at *m/z* 290.1387 and 364.1907 indicating that hydroxylation occurred in desmethyldonepezil. A benzylic carbocation (*m/z* 91.0547) peak was also observed, suggesting that there was no modification of the benzyl moiety. The more polar metabolites **M7a** (5.41 min) and **M7b** (5.76 min) exhibited shorter retention times than that of the relatively less polar piperidine ring-hydroxylated donepezil (**M3a**, 6.85 min). **M7a** and **M7b** were tentatively identified as 6-*O*- and 5-*O*-desmethyl-piperidine ring-hydroxylated donepezil, respectively, because 6-*O*-desmethyldonepezil (**M1**, 7.64 min) eluted slightly faster than 5-*O*-desmethyldonepezil (**M2**, 7.85 min). Metabolite **M7c** had major fragment ions that were 14 Da smaller than those of benzene ring-hydroxylated donepezil at *m/z* 137.0594, 259.1314, and 276.1595 (Appendix A), suggesting that one methyl group was removed from the dimethoxyindanone moiety of hydroxydonepezil. Similar to **M3c**, a benzylic carbocation was observed at *m/z* 107.0495 in the MS/MS spectrum of **M7c**. The more polar metabolite **M7c** (6.11 min) had a shorter retention time than that of the relatively-less polar **M3c** (7.59 min). Based on these observations, metabolite **M7c** was identified as *O*-desmethyl-benzene ring-hydroxylated donepezil.

Metabolite **M7d** and **M7e** had major fragment ions that were 14 Da smaller than those of 3-hydroxydonepezil at *m/z* 272.1281, 290.1387, and 364.1907 (theoretical *m/z*, Appendix A), suggesting that one methyl group was removed from the dimethoxyindanone moiety of hydroxydonepezil. In addition, their fragment ions were 16 Da larger than those of desmethyldonepezil at *m/z* 245.1172, 290.1387, and 364.1907 indicating that hydroxylation occurred in desmethyldonepezil. A benzylic carbocation (*m/z* 91.0542) peak was also observed, suggesting that there was no modification of the benzyl moiety. The more polar metabolites **M7d** (6.51 min) and **M7e** (6.76 min) exhibited shorter retention times than that of the relatively less polar 3-hydroxydonepezil (**M3b**, 7.35 min). **M7d** and **M7e** were tentatively identified as 6-*O*- and 5-*O*-desmethyl-3-hydroxydonepezil, respectively, because 6-*O*-desmethyldonepezil (**M1**, 7.64 min) eluted slightly faster than 5-*O*-desmethyldonepezil (**M2**, 7.85 min).

The metabolites **M8a** (8.39 min) and **M8b** (8.68 min) had longer retention times than desmethyldonepezil (**M1** and **M2**, 7.64~7.85 min), which was indicative of *N*-oxide formation and showed characteristic fragment ions at *m/z* 91.0542, 137.0597, 229.1223, and 274.1438 (Appendix A), similar to the fragmentation pattern of donepezil *N*-oxide (**M4**). The annotation of metabolites **M8a** (6-*O*-desmethyldonepezil *N*-oxide) and **M8b** (5-*O*-desmethyldonepezil *N*-oxide) was deduced by comparing the retention times with those of metabolites **M1** and **M2**.

The protonated molecular ions of metabolites **M9a** and **M9b** were observed at *m/z* 276.1594 (theoretical *m/z*) with mass errors of under 3.5 ppm, and they eluted at 3.17 and 3.40 min, respectively. The fragment ions were the same as those of metabolites demethylated from the dimethoxyindanone moiety of **M5** at *m/z* 137.0597, 175.0754, 191.0703, and 258.1489 (theoretical *m/z*, Appendix A). In addition, the fragment ion was not detected at *m/z* 91.0542, corresponding to the benzyl group and the characteristic fragment ion generated from loss of the benzyl group in benzyl-tetrahydropyridine was observed at *m/z* 82.0651 (theoretical *m/z*), similar to that of **M5**. This indicates that both *O*-demethylation from the dimethoxyindanone moiety and *N*-debenzylation from the piperidine ring occurred. The annotation of metabolites **M9a** (6-*O*-desmethyl-M5) and **M9b** (5-*O*-desmethyl-M5) was deduced by comparing the retention times with those of metabolites **M1** and **M2**.

The protonated molecular ions of metabolites **M10a** and **M10b** were observed at *m/z* 412.2118 (theoretical *m/z*), and they eluted at 6.00 and 8.16 min, respectively. The accurate mass measurements indicated that the chemical formula is C_24_H_29_NO_5_, suggesting that two oxygen atoms were introduced into donepezil. Fragment ions of metabolite **M10a** were observed at *m/z* 288.1608 and 394.2017, and those of metabolite **M10b** were observed at *m/z* 286.1436 and 304.1544. A benzylic carbocation (*m/z* 91.0547) peak was also observed, suggesting there was no modification of the benzyl moiety. These data indicate that dihydroxylation occurred, but the exact location could not be determined. However, **M10a** may not contain aromatic hydroxylation and *N*-oxidation because a fragment ion (*m/z* 394.2017) produced from the loss of a water molecule was found in the MS/MS spectrum of **M10a**. Both the aromatic ring-hydroxylated metabolite and *N*-oxide showed only a small amount of water loss in the MS/MS spectra [38]. Metabolite **M10b** may have an *N*-oxide structure within the compound because it (8.16 min) had a longer retention time compared with the three hydroxydonepezils (**M3a**, **M3****b**, and **M3c**; 6.88~7.60 min), which is characteristic of *N*-oxide metabolites of other drugs [7,39].

The protonated molecular ion of metabolites **M11a** and **M11b** were observed at *m/z* 306.1700 (theoretical *m/z*) and eluted at 2.64 and 3.42 min, respectively. The accurate mass measurements indicated that the chemical formula is C_17_H_23_NO_4_, suggesting that one oxygen atom was introduced into *N*-desbenzyldonepezil. Fragment ions of metabolite **M11** were observed at *m/z* 259.1329, and 288.1594 (added oxygen to *m/z* 243.1380 and 272.1645), which was the same as **M5** at *m/z* 82.0651, 151.0754, 189.0910, and 205.0859. Though a 4-methyl-hydroxypiperidine ion was not found in the MS/MS spectra of **M11**, the fragment ion at *m/z* 96.0808 was produced from the loss of water molecule from 4-methyl-hydroxypiperidine, suggesting that piperidine ring hydroxylation occurred. However, the exact hydroxylation sites on the piperidine ring for metabolites **M11a** and **M11b** could not be determined.

### 3.2. Identification of Donepezil Reactive Metabolites

Glutathione and potassium cyanide adducts were used to trap reactive intermediates of donepezil and its metabolites in liver microsomal fractions [40]. No glutathione adducts were detected following incubation with liver microsomes and glutathione. Donepezil contains an alicyclic tertiary amine group, *N*-benzyl piperidine, that is able to form reactive iminium intermediates, which can be captured using KCN. Only one cyanide adduct (**M12**) was observed only in rat liver microsomal incubation samples (Figure 5a). The protonated molecular ion of the cyanide adduct was observed at *m/z* 315.1703 (theoretical *m/z*) and eluted at 5.68 min. An MS/MS scan yielded a molecular ion peak at *m/z* 315.1703 (Figure 5a) and a characteristic fragment ion at *m/z* 96.0811, 151.0752, 189.0906, 243.1377, 270.1487, and 288.1591 (Figure 5b). The fragment ion at *m/z* 288.1591 may represent the loss of the hydrogen cyanide molecule, which is characteristic of the cyanide adduct [40,41]. Metabolic reactions for α-cyano *N*-desbenzyldonepezil (**M12**) may represent *N*-debenzylation and the addition of a cyano group to the piperidine ring; therefore, this is the first report of an *N*-desbenzyldonepezil cyanide adduct. Evidence linking the bioactivation of an alicyclic amine into a reactive iminium ion after *N*-dealkylation was previously reported for vandetinib, a piperidine-containing drug [42]. A bioactivation pathway for donepezil was proposed (Figure 6). The *N*-benzyl piperidine ring in donepezil underwent P450-catalyzed *N*-debenzylation, oxidation and subsequent dehydration to form an imine, which was trapped by potassium cyanide to form a stable adduct. 

The production of reactive metabolites is believed to be a major mechanism for the side effects of drugs, especially idiosyncratic drug-induced liver injury (DILI) [43,44]. Although infrequent liver injury related to donepezil has been reported post-marketing, the mechanism of toxicity is not yet well understood. As previously reported, alicyclic amines are oxidized by cytochrome P450 and other drug-metabolizing enzymes to iminium ions [45,46]. Furthermore, iminium ions are strong nucleophiles that can be covalently bonded to proteins or deoxyribonucleic acid bases and are known to mediate many toxicities [47,48]. These reactive iminium intermediates may be associated with the reported side effects of donepezil. However, reactive metabolite formation does not necessarily trigger hepatotoxicity; therefore, to account for the role of these potential reactive metabolites, the relationship between donepezil’s reactive metabolites and donepezil-induced hepatotoxicity should be further evaluated.

## 4. Conclusions

In this study, the in vitro metabolism of donepezil by HLM, MLM, and RLM was examined using a non-targeted metabolomics approach combined with molecular networking using LC-MS/MS. The structure of the metabolites was characterized based on accurate mass and fragmentation patterns. A total of 21 donepezil metabolites (17 metabolites in HLM, 21 metabolites in MLM, and 17 metabolites in RLM) were detected, including those previously reported. A total of 14 Metabolites (**M3a**, **M3b**, **M7a**, **M7b**, **M7d**, **M7e**, **M8a**, **M8b**, **M9a**, **M9b**, **M10a**, **M10b**, **M11a**, and **M11b**) were newly identified. Donepezil was metabolized in various ways after incubation with liver microsomes. The results showed that metabolites are formed through four major metabolic pathways. (1) *O*-demethylation, (2) hydroxylation, (3) *N*-oxidation, and (4) *N*-debenzylation. Twenty-one Phase I donepezil metabolites in addition to one potential reactive metabolite were detected, and the position of the metabolic reaction is proposed in Figure 7. The proposed metabolic pathway is also depicted in Figure 8. One potential reactive metabolite (cyano adduct) was observed in rat liver microsomal incubation samples for the first time.

In conclusion, our results indicate that a non-targeted metabolomics approach combined with molecular networking is a reliable tool to identify and detect unknown drug metabolites. We newly identified 14 metabolites with this method. The in vitro metabolite profiling of donepezil identified new metabolites and revealed a differential donepezil metabolism in HLM, MLM, and RLM. A total of 21 donepezil metabolites generated through various metabolic reactions were successfully characterized. In rat liver microsomal incubation samples, for the first time, we identified a reactive metabolite in the form of a cyanide adduct of *N*-desbenzyldonepezil. Metabolomics and molecular networking approaches will be widely used to identify novel metabolites of drug.

## Figures and Tables

**Figure 1 pharmaceutics-13-00936-f001:**
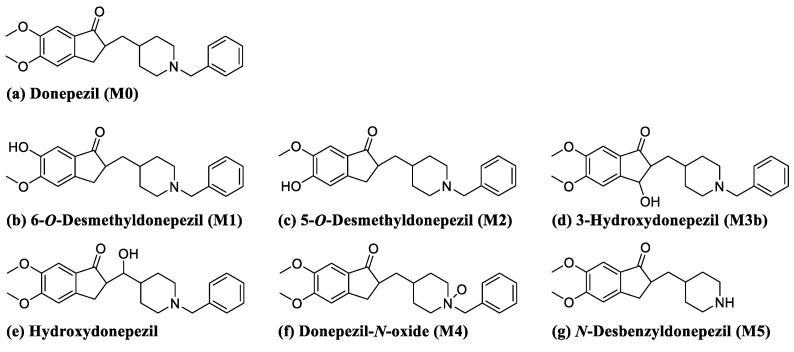
Chemical structures of donepezil and its commercially available metabolites.

**Figure 2 pharmaceutics-13-00936-f002:**
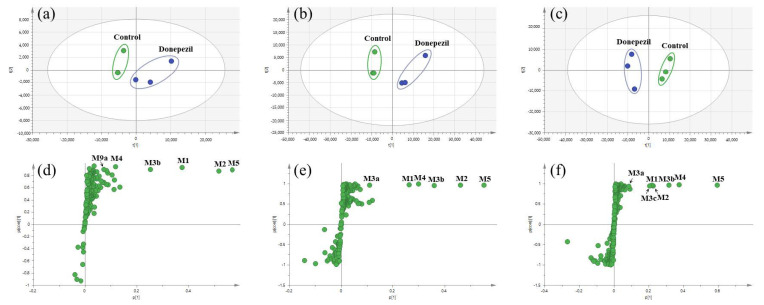
Multivariate analysis of donepezil metabolites in human liver microsomes (HLM) (**a**), mouse liver microsomes (MLM) (**b**), and rat liver microsomes (RLM) (**c**). Control groups were incubated in the absence of nicotineamide adenine dinucleotide phosphate reduced form (NADPH), and donepezil groups were incubated in the presence of NADPH. Score plots were generated by a principal component analysis based on the liquid chromatography–high resolution mass spectrometry data of human liver microsomes (HLM) (**a**), mouse liver microsomes (MLM) (**b**), and rat liver microsomes (RLM) (**c**). Loading S−plots were generated by an orthogonal partial least square−discriminant analysis (OPLS−DA) based on the liquid chromatography–high resolution mass spectrometry data of HLM (**d**), MLM (**e**), and RLM (**f**). The *p* [1] values represent the relevant abundance of ions, and *p*(corr) [1] values represent the interclass difference. Green dots on the S−plot represent variables (mass spectral data) which reflect the influence of each variable in the two groups (control vs. donepezil). Variables that are farthest from the origin in the S−plot are selected as potential donepezil metabolites. Identified metabolites were marked with abbreviations on the S−plot, and metabolite lists are given in Table 1. Data processing and model construction are described in the Materials and Methods.

**Figure 3 pharmaceutics-13-00936-f003:**
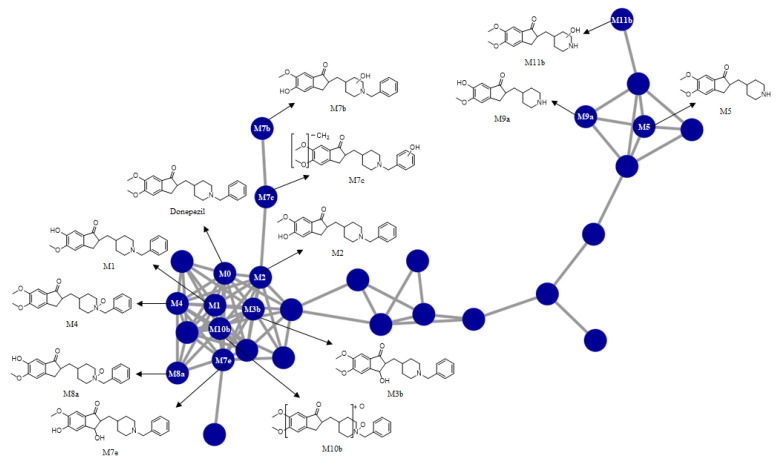
Representative molecular network of the MS/MS spectra obtained by the liquid chromatography–high resolution mass spectrometry analysis of the liver microsomal incubation mixtures of donepezil in the presence of nicotineamide adenine dinucleotide phosphate reduced form. Unlabeled blue circles are variables that are networked with donepezil but were confirmed not to be metabolites of donepezil based on MS/MS spectral analysis.

**Figure 4 pharmaceutics-13-00936-f004:**
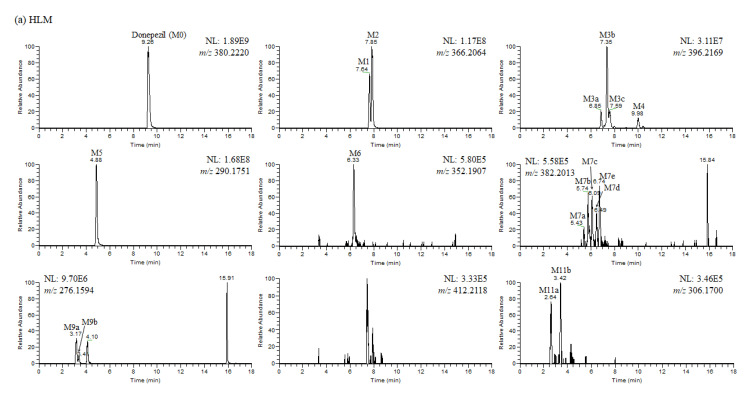
(**a**) Representative extracted ion chromatograms of donepezil and its metabolites obtained from the liquid chromatography–high resolution mass spectrometry analysis of human liver microsomes (HLM) incubation mixtures of donepezil in the presence of nicotineamide adenine dinucleotide phosphate reduced form (1 h, 37 °C). NL stands for normalization level, which describes the intensity of the highest peak in each chromatogram. (**b**) Representative extracted ion chromatograms of donepezil and its metabolites obtained from the liquid chromatography–high resolution mass spectrometry analysis of mouse liver microsomes (MLM) incubation mixtures of donepezil in the presence of nicotineamide adenine dinucleotide phosphate reduced form (1 h, 37 °C). NL stands for normalization level, which describes the intensity of the highest peak in each chromatogram. (**c**) Representative extracted ion chromatograms of donepezil and its metabolites obtained from the liquid chromatography–high resolution mass spectrometry analysis of rat liver microsomes (RLM) incubation mixtures of donepezil in the presence of nicotineamide adenine dinucleotide phosphate reduced form (1 h, 37 °C). NL stands for normalization level, which describes the intensity of the highest peak in each chromatogram.

**Figure 5 pharmaceutics-13-00936-f005:**
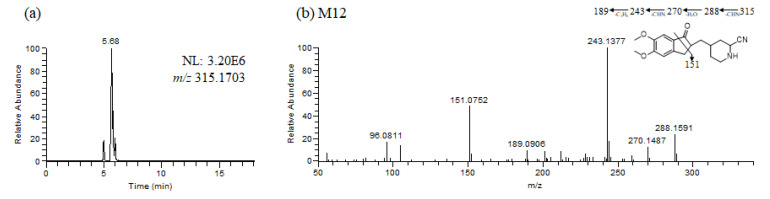
Representative extracted ion chromatograms (**a**), product ion scan mass spectrum and proposed fragmentation scheme (**b**) of reactive metabolites produced from the incubation of donepezil with rat liver microsomes in the presence of nicotineamide adenine dinucleotide phosphate reduced form and potassium cyanide (1 h, 37 °C).

**Figure 6 pharmaceutics-13-00936-f006:**
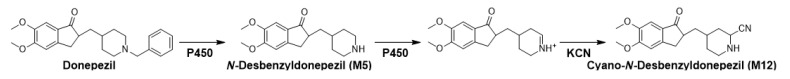
Proposed bioactivation mechanism of the piperidine ring of donepezil and the formation of a cyano adduct.

**Figure 7 pharmaceutics-13-00936-f007:**
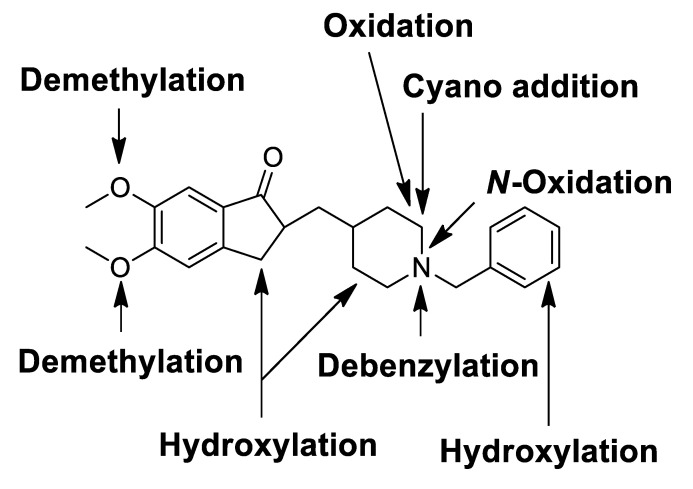
Chemical structure of donepezil showing locations of Phase I metabolic reactions and the bioactivation pathway.

**Figure 8 pharmaceutics-13-00936-f008:**
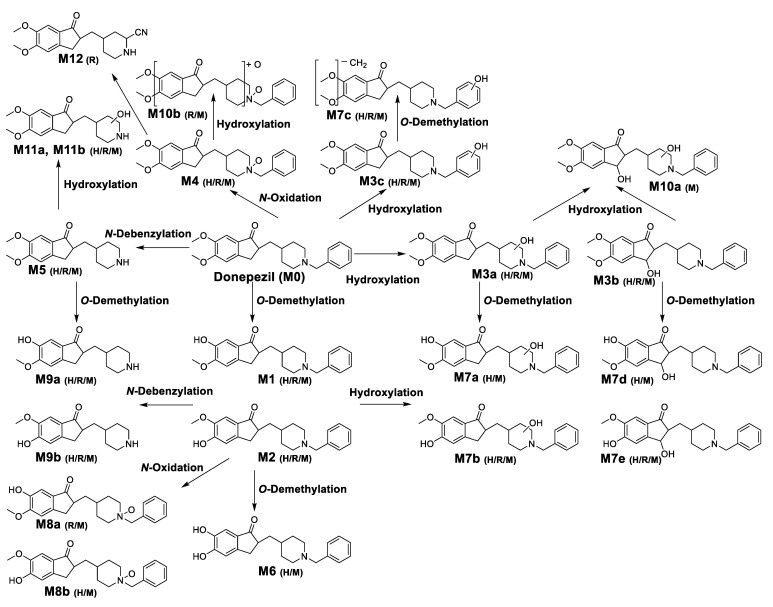
Proposed metabolic pathways of donepezil in human liver microsomes (H), rat liver microsomes (R), and mice liver microsomes (M).As a result of comparing metabolites between species, donepezil had similar metabolic profiles in HLM, MLM, and RLM. *N*-Desbenzydonepezil (**M5**) was the most abundant metabolite in all incubations. Meier-Davis et al., (2012) also reported *N*-desbenzydonepezil as the most abundant metabolite in rat and human plasma following oral administration [37] and in rat plasma, liver, and kidney after a single oral administration of donepezil [25]. The metabolite **M5** was further metabolized through *O*-demethylation (**M9**) or hydroxylation (**M11**). The second most abundant metabolite was *O*-desmethyldonepezil (**M1** and **M2**) following HLM and MLM incubation, which was further metabolized by hydroxylation to form additional related metabolites (**M7a**–**M7e**) compared with RLM. In contrast, donepezil *N*-oxide (**M4**) was a much more abundant metabolite in MLM and RLM incubations compared with that of HLM. In addition, dihydroxydonepezil (**M10**) was characteristically observed only in MLM and RLM incubations.

**Table 1 pharmaceutics-13-00936-t001:** Summary of donepezil metabolites produced through in vitro metabolism by liver microsomes.

Metabolites	t_R_ (min)	[M+H]^+^		Error (ppm)	Formula	Fragment Ions (*m/z*)	Reaction Type	Source		
		Measured	Theoretical					HLM	MLM	RLM
Donepezil	9.29	380.2214	380.2220	−1.58	C_24_H_29_NO_3_	91.0546, 151.0751, 172.1115, 189.0909, 205.0862, 243.1378, 273.1484, 288.1591, 362.2105				
M1	7.64	366.2063	366.2064	−0.27	C_23_H_27_NO_3_	91.0548, 137.0599, 175.0753, 191.0701, 229.1226, 259.1338, 274.1441, 348.1962	6-*O*-demethylation	+	+	+
M2	7.88	366.2065	366.2064	0.27	C_23_H_27_NO_3_	91.0548, 137.0596, 175.0757, 191.0710, 229.1223, 259.1328, 274.1437, 348.1959	5-*O*-demethylation	+	+	+
M3a	6.88	396.2171	396.2169	0.50	C_24_H_29_NO_4_	91.0548, 151.0754, 189.0918, 205.0860, 259.1331, 286.1438, 304.1552, 378.2061	hydroxylation	+	+	+
M3b	7.40	396.2174	396.2169	1.26	C_24_H_29_NO_4_	91.0548, 172.1122, 205.0867, 259.1335, 286.1440, 304.1541, 378.2064	hydroxylation	+	+	+
M3c	7.60	396.2177	396.2169	2.02	C_24_H_29_NO_4_	107.0495, 151.0758, 189.0915, 205.0864, 243.1397, 273.1475, 290.1750, 378.2065	hydroxylation	+	+	+
M4	10.06	396.2163	396.2169	−1.51	C_24_H_29_NO_4_	91.0547, 151.0752, 189.0912, 205.0860, 243.1389, 273.1495, 288.1592, 304.1579, 378.2078	*N*-oxidation	+	+	+
M5	4.89	290.1750	290.1751	−0.34	C_17_H_23_NO_3_	82.0657, 151.0752, 189.0909, 205.0859, 243.1392, 272.1637, 273.1481	*N*-debenzylation	+	+	+
M6	6.34	352.1905	352.1907	−0.57	C_22_H_25_NO_3_	91.0547, 123.0441, 215.1069, 245.1170, 260.1281, 334.1802	didemethylation	+	+	-
M7a	5.41	382.2012	382.2013	−0.26	C_23_H_27_NO_4_	91.0547, 290.1377, 364.1908	*O*-demethylation + hydroxylation	+	+	-
M7b	5.76	382.2009	382.2013	−1.05	C_23_H_27_NO_4_	91.0547, 272.1277, 290.1386, 364.1899	*O*-demethylation + hydroxylation	+	+	+
M7c	6.11	382.2003	382.2013	−2.62	C_23_H_27_NO_4_	107.0495, 137.0594, 175.0757, 191.0706, 258.1484, 259.1314, 276.1595, 364.1914	*O*-demethylation + hydroxylation	+	+	+
M7d	6.51	382.2015	382.2013	0.52	C_23_H_27_NO_4_	91.0547, 245.1186, 272.1271, 290.1386, 364.1927	*O*-demethylation + hydroxylation	+	+	-
M7e	6.76	382.2012	382.2013	−0.26	C_23_H_27_NO_4_	91.0548, 153.0547, 191.0700, 207.0652, 245.1174, 275.1276, 290.1393, 364.1907	*O*-demethylation + hydroxylation	+	+	+
M8a	8.41	382.2001	382.2013	−3.14	C_23_H_27_NO_4_	91.0547, 137.0598, 229.1223, 274.1437	*O*-demethylation + *N*-oxidation	-	+	+
M8b	8.69	382.2005	382.2013	−2.09	C_23_H_27_NO_4_	91.0547, 137.0598, 229.1223, 274.1439	*O*-demethylation + *N*-oxidation	-	+	+
M9a	3.18	276.1599	276.1594	1.81	C_16_H_21_NO_3_	82.0657, 137.0599, 175.0753, 191.0702, 229.1225, 258.1487, 259.1321	*O*-demethylation + *N*-debenzylation	+	+	+
M9b	3.41	276.1585	276.1594	−3.26	C_16_H_21_NO_3_	82.0657, 137.0597, 175.0754, 191.0703, 229.1229, 258.1494, 259.1318	*O*-demethylation + *N*-debenzylation	+	+	+
M10a	6.00	412.2111	412.2118	−1.70	C_24_H_29_NO_5_	91.0547, 288.1608, 394.2017	dihydroxylation	-	+	-
M10b	8.16	412.2138	412.2118	4.85	C_24_H_29_NO_5_	91.0547, 203.0705, 243.1022, 286.1436, 304.1544, 394.2024	hydroxylation + *N*-oxidation	-	+	+
M11a	2.66	306.1698	306.1700	−0.33	C_17_H_23_NO_4_	84.0813, 151.0753, 189.0909, 205.0857, 243.1383, 259.1332, 288.1593	hydroxylation + *N*-debenzylation	+	+	+
M11b	3.45	306.1699	306.1700	−0.33	C_17_H_23_NO_4_	82.0657, 151.0753, 189.0909, 205.0857, 243.1380, 259.1334, 288.1597	hydroxylation+ *N*-debenzylation	+	+	+
M12	5.67	315.1701	315.1703	−0.63	C_18_H_22_N_2_O_3_	151.0752, 189.0907, 243.1378, 270.1480, 288.1597	cyanide adduct	-	-	+

## Data Availability

All data in this study have been included in this manuscript.

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
