# Peer review of "In Vitro Metabolism of Donepezil in Liver Microsomes Using Non-Targeted Metabolomics"

_pharmaceutics, 2021, doi:10.3390/pharmaceutics13070936_

Round 1

Reviewer 1 Report

In this paper, authors investigate metabolism of the drug donepezil using untargeted LC-MS/MS approach which allows discovery of various metabolites of the drug including novel metabolites. The paper presents an elegant way of combining spectral data and molecular networking to identify unknown metabolites through MS/MS fragmentation similarity to parent drug. The analysis of the metabolism of donepezil and identification of the total of 21 metabolites was performed across different species - in human, mouse, and rat liver microsomes.

All proposed structures of metabolites are supported by interpretation of experimental MS/MS spectra in cases when the authentic chemical standards were not available for drug metabolites. In addition, authors suggest an explanation of the mechanism of liver damage induced by this drug through the production and analysis of reactive metabolite (cyano adduct).

The paper is generally well-written, is sufficiently descriptive, it contains adequate citations, and the quality of the figures is good. In my opinion, this paper provides important information, more specifically regarding donepezil metabolism, and more generally presents an efficient metabolomics approach to study drug metabolism and discovery of unknown metabolites.  I strongly recommend to publish the paper as is.

Author Response

Thank you for your recommendation. 

Reviewer 2 Report

  • Overall statement:

The article aimed to do a metabolomics study to unravel metabolites of Donepezil which is a known cholinesterase inhibitor, and has a used a new method including liver microsomes, and molecular networking. The authors have found new metabolites and a unique metabolite in one of the samples, which is the rat liver microsomes.

  • Overall ​article strengths:​

The manuscript used a new method to detect new metabolites of Donepezil, and the method is promising, also it is generally well written, the experiment is sound and the references are relevant.

  • Major points:
  • Page 2: introduction: line 21 “…tandem MS data” , is missing a reference. Lines 24, 25: are missing references. Line 27 is missing a reference.
  • Line 27 page 2: replace “this medication” with the drug’s name: Donepezil.
  • Page 2: Line 39: missing the references of the 2 mentioned studies.
  • In the introduction: it is best to mention the aim clearly and strongly, as well as adding more information about the published data on Donepezil metabolism.
  • Are the liver microsomes having the same quality and quantity of the enzymes? A statement commenting on the liver microsomes is required, as they are compared to each other in the manuscript.
  • Page 3: add a statement about Donepezil, as to how it was prepared, also for the reduced NAD.
  • Page 3 line 10: “microsomal protein” to be corrected to be: microsomal proteins.
  • Page 4: Line 17: statements mentioning how the statistics were done i.e. the specifics of the PCA and OPLS-DA plots, what was the input exactly, are required.
  • Page 4: Line 46: for the rest of the metabolites, whose structures were not confirmed, they are mentioned in Figure 8, are they not confirmed? A statement is required to clarify whether the rest of the structures are confirmed or yet to be confirmed.
  • Page 5: last sentence, with the reference [36], is the MS/MS spectrum of Donepezil included in the manuscript? If yes, the figure number is best to be included, if not, then the figure is best to be included and cited.
  • Page 6: Figure 2 is relatively small and the labeled dots are hard to be read, they can possibly be enlarged, or the labeled dots mentioned in the legend. The legend is generally very concise and more elaboration is required: control: absence of reduced NAD and the Donepezil: addition of the reduced NAD… Again, the PCA and OPLS-DA plots were not clear as to how they were generated. For the unlabeled green dots on the OPLS-DA plot, are they referring to uncharacterized metabolites? It is best to be mentioned in the legend to be understood easier.
  • Page 8: Figure 4A, 4B, 4C: the legends are not clear as to which panel is with reduced NAD and which is without, and there are several acronyms that can be included in the legend, also the panels are overall unlabeled, small and the letters are somewhat hard to read, it is best if they could be enlarged.
  • Table 1: title best to be modified along the lines of: “produced through/via in vitro metabolism by liver microsomes”, as a linking word is missing, as well as details on the in vitro method.
  • Page 13: Did the authors submit the data to GNPS platform or PubChem or any other repository to be deposited and available later on?
  • Supplementary figure: the figures need to be larger, each panel labeled, the acronyms to be explained, and to be mentioned as to which figure is for which metabolite.
  • Page 15: Line 5: the figure needs to be included in the supplementary data, as to the authentic standard.
  • Page 17: Line 10 and figure 5: the last peak mentioned was not in the figure 5b (m/z 315). Figure 5a has it, it is best to be mentioned in the sentence in line 10.
  • Page 18: Line 7: it is best to mention all the new metabolites, to be more thorough, instead of mentioning examples.
  • Page 20: A paragraph or statement mentioning possible caveats of the experiment is required, e.g. are the microsomes components exactly the same quantity and quality, and how to make it more accurate?! The paragraph also is best to be including the future of this study and to highlight its importance. The manuscript has mentioned that this method is reliable, how does it compare to other methods, e.g. what was the percentage of new metabolites that were identified by this method?
  • Minor points:
  • Title: needs to be a bit more detailed and include something interesting about the results and omit the word “study”.
  • Page 19: Figure 8: M11a, M11b, M12 are missing from the figure but included in the table, is there a certain rationale for it? Also, it is best to add M0 on Donepezil inn Figure 8.
  • Page 18: Line 9: correction for “liver microsome incubation” to be: after incubation with liver microsomes, to be clearer to the readers.
  • Page 15: Line 31: hyphen should be removed to be: “relatively less”.
  • Page 15: Line 51: “suggesting that”.
  • Page 15: Line 54: for M7d and M7e, were they confirmed by MS/MS? A clarification for how their structures were elucidated, is best to be added.
  • Page 16: Line 32: best to replace “impossible” to be: could not be determined.
  • Page 16: Line 52: to correct: “was occurred” to had occurred or occurred.
  • Page 17: Line 2: a reference is required.
  • Page 7 figure 3: are the unlabeled blue circles metabolites that were found but their structures were not elucidated? A statement explaining them shall make it easier to be understood.

Author Response

We revised our manuscript based on your comments.
Please look over the attached.

Round 2

Reviewer 2 Report

There are just now a few grammatical and format typos that require minor proofreading: e.g. section 2.5 is all italic, it needs to be fixed, and Table 1 title: "via" to be written instead of "through". Please revise formatting and grammar throughout the manuscript. Otherwise, it is acceptable.

Author Response

We revised our manuscript based on your comments.
